# Spatio-Temporal Correlation Between Radon Emissions and Seismic Activity: An Example Based on the Vrancea Region (Romania)

**DOI:** 10.3390/s25030933

**Published:** 2025-02-04

**Authors:** David Montiel-López, Sergio Molina, Juan José Galiana-Merino, Igor Gómez, Alireza Kharazian, Juan Luís Soler-Llorens, José Antonio Huesca-Tortosa, Arianna Guardiola-Villora, Gonzalo Ortuño-Sáez

**Affiliations:** 1Multidisciplinary Institute for Environmental Studies, University of Alicante, Ctra. San Vicente del Raspeig, s/n, 03080 Alicante, Spain; sergio.molina@ua.es (S.M.); igor.gomez@ua.es (I.G.); alireza.kharazian@gcloud.ua.es (A.K.); 2Department of Applied Physics, University of Alicante, Ctra. San Vicente del Raspeig, s/n, 03080 Alicante, Spain; 3Department of Physics, Systems Engineering and Signal Theory, University of Alicante, Ctra. San Vicente del Raspeig, s/n, 03080 Alicante, Spain; jj.galiana@ua.es; 4University Institute of Physics Applied to Sciences and Technologies, University of Alicante, Ctra. San Vicente del Raspeig, s/n, 03080 Alicante, Spain; 5Department of Earth and Environmental Sciences, University of Alicante, Ctra. San Vicente del Raspeig, s/n, 03080 Alicante, Spain; jl.soler@ua.es; 6Department of Architectural Constructions, University of Alicante, Ctra. San Vicente del Raspeig, s/n, 03080 Alicante, Spain; ja.huesca@ua.es; 7Department of Continuum Mechanics and Theory of Structures, Universitat Politècnica de València, Camino de Vera, s/n, 46022 Valencia, Spain; aguardio@mes.upv.es; 8Independent Researcher, 03300 Orihuela, Spain; gosaez@orihuela.es

**Keywords:** radon, seismicity, correlation, lag, signal, earthquake, Vrancea, Romania

## Abstract

Radon gas anomalies have been investigated as potential earthquake precursors for many years. In this work, we have studied the possible correlations between radon emissions and the seismic activity rate for a given region to test if the existing correlation may be later used to forecast the occurrence of earthquakes larger than a given magnitude. The Vrancea region (Romania) was chosen as a study area since it is being surveilled by a multidisciplinary real-time monitoring network, and at least seven earthquakes with magnitudes greater than 4.5 Mw have occurred in this area in the period from 2016 to 2020. Our research followed several steps: First, the recorded radon signals were preprocessed (detrended, deseasoned and smoothed). Then, the station’s signals were correlated in order to check which stations are recording radon anomalies due to the same regional tectonic process. On the other hand, the seismic activity rate was computed using the earthquakes in the main catalogue of the region that are able to generate radon emissions and can be registered at several stations. The obtained results indicate a significant correlation between the seismic activity rate and the temporal series of radon anomalies. A temporal lag between the seismic activity rate and the radon anomalies was found, which can be related to the proximity to the epicentre of the main earthquake in each of the studied subperiods. Changes in the regional tectonic stress field could explain why the seismic activity rate and radon anomalies are correlated over time. Further research could focus on obtaining a function to forecast the seismic activity rate using the following as dependent variables: the radon anomalies recorded at several stations, the distance from the stations, and tectonic factors such as the fault system, azimuth, type of soil, etc.

## 1. Introduction

The interest in radon as a precursor signal candidate in earthquake forecasting has been increasing since the late XX century, especially since 1980, as thorough research has examined cases where correlations between radon anomalies and earthquakes have been found [1,2,3,4]. Although the use of this chemical species as an earthquake precursor has not been disregarded, there has been a shift in the mindset regarding the possibility of earthquake prediction. The complexity of the processes that occur in the lithosphere (from which, in most cases, only indirect measurements can be obtained) and the impossibility of scaling up laboratory experiments limit the sources for a physiochemical model that accurately describes radon anomalies in relation to the earthquake preparation process. These are some of the difficulties regarding earthquake prediction based on radon signal anomalies. Nevertheless, by relying on statistics and adopting a regional approach, rather than a global scope [5], the use of radon as a seismicity indicator could still prove to be useful.

In this sense, ^222^*Rn* soil gas has been monitored in some regions, with the aim of relating it with the occurrence of earthquakes. For instance, in shallow-seismicity regions around the Black Sea, Nevinksy et al. [6] found that radon levels were anomalous before regional earthquakes and returned to normal 1–2 days after they happened, and Piersanti et al. [7] correlated temperature with radon and found a slight peak 30–40 days before three significant earthquakes. These works have highlighted the importance of expanding the radon-monitoring networks. In intermediate-to-deep seismicity regions, the case of the Vrancea region (Romania) can be cited, where Toader et al. [8] analysed the results of the multi-parametric station network located along some of the main faults of the region and identified radon anomalies happening from 10 to 37 days before major earthquakes in the period of 2018–2020. They also designed a logic tree to use several geophysical and geochemical parameters to enable Operational Earthquake Forecasting (OEF from now on) in the region. Galiana-Merino et al. [9] also studied the correlation between the radon residual at one of the recording stations in the region and the seismic activity rate using Discrete Wavelet Transform (DWT) analysis [10] for the period from December 2015 to July 2020 and found similarities between the signals. They found similarities between both time series that could enable the estimation of the seismic activity rate using the radon signal, which, in turn, could be useful regarding Time-Dependent Probabilistic Seismic Hazard Analysis (TD-PSHA).

In this work, we revisit the Vrancea region in Romania to study the correlation between the radon signals of different multi-parametric stations located in the area of study in order to analyse its dependence on the seismic activity rate. A description of the area of study in terms of its seismicity is presented, as this will help filter the earthquakes which may cause the radon emissions recorded at each station.

The structure of this study is as follows: we first present an introduction to the multi-parametric network and the seismicity in Vrancea (Romania) as well as the radon signal; then, we detail the cross-correlation between the radon station signals; following this, we detail the seismic activity rate computation for each of the correlated stations; we then present the correlation study for the radon and seismic activity rate time series; and finally, we present a discussion of the main results.

## 2. The Vrancea Region (Romania): A Multi-Parametric Network

### 2.1. Description of the Network

The Vrancea region in Central–Eastern Romania has been extensively monitored to determine a range of geophysical and geochemical parameters in relation to the intermediate-to-deep seismicity characteristic of this area. The National Institute for Earth Physics (NIEP) of Romania developed a multidisciplinary network for monitoring these parameters: AeroSolSys [11]. Gas concentrations (CO_2_, ^222^*Rn*), temperature, wind and pressure, humidity and local magnetic field variations, among other parameters, are measured by the deployed network.

The main objective of the AeroSolSys project is to provide authorities with updated information on these parameters regarding the existing seismic hazards in the region.

The parameters are studied in a time-dependent framework and correlated with seismicity considering their variations and their dependence on external factors (such as temperature, humidity, proximity to water courses, etc.), using extended time periods to ensure a correct analysis.

### 2.2. Seismicity and Seismotectonics

The seismicity in Romania can be described as bimodal in terms of the epicentre’s depth distribution. In the case of Vrancea, the most damaging earthquakes have occurred in the intermediate-to-deep part of the upper lithosphere (from 60 to 180 km in depth), where the main seismic activity is clustered [12,13].

The Vrancea region’s seismicity can be studied using NIEP’s ROMPLUS catalogue (https://data.mendeley.com/datasets/tdfb4fgghy/2, accessed on 20 February 2024) [14,15,16,17], which was downloaded in order to represent the regional seismicity. The catalogue for the Vrancea region (constrained in this work by longitudes from 25.8° E to 27.5° E and the latitudes of 45.0° N and 46.2° N) spans from the year 984 to 2023 with a magnitude range from 0 to 7.9 Mw and depths from surface to 218.4 km, with a total of 14,291 events. It is represented in Figure 1.

It can be seen that the most intense earthquakes are concentrated around the eastern-most part of the Vrancea region towards the north-eastern limit of the Buzau region. Regarding the origin of seismicity, one of the hypotheses is based on the presence of a subducted oceanic slab. Nevertheless, Knapp et al. 2005 [18] argued that a delamination process could explain the seismicity in the foreland of the Carpathian system not accounted for in earlier models for the subducted oceanic slab scenario. More recently, Müller et al. 2010 [19] pointed towards a non-coupled (or detached) slab scenario (with respect to the upper crust). In contrast to this model, Petrescu et al. 2021 [20] proposed a weakly coupled slab scenario for Vrancea compatible with the observed foreland deformation. In terms of mechanisms that explain the physiochemical processes involved in seismicity, Ferrand and Manea (2021) [21] found a correlation between the intermediate–deep seismicity of the Vrancea region and the dehydration of the minerals in the subducted oceanic slab in this region.

The most intense earthquakes (Mw ≥ 4.5) during the last 10 years in the area of study are listed in Table 1.

### 2.3. Radon Monitoring

In the case of the Vrancea region, from earlier than 2016 onwards, the Radon Monitoring Network has been recording ^222^*Rn* signals along with other parameters such as temperature, CO_2_ concentration, pressure, humidity, etc.

The Radon Monitoring Network is composed of Radon Scout Plus stations (https://sarad.de/product-detail.php?lang=en_US&cat_ID=2&p_ID=37, accessed on 22 January 2025), which provide relatively accurate and precise radon measurements, with a typical instrumental error lower than 6%, depending on environmental factors and proper use. Each data point (for each minute) is measured over integration periods of 3 h for which the statistical error is around ±20%. Regular calibration and appropriate measurement conditions are essential to minimise sources of error, including environmental interference, sensor drift, and the influence of radon decay products.

More details about the radon stations and the monitoring network can be found in the references [8,22,23].

For this work, we selected a set of indoor radon stations using the location and data availability as filters (Table 2).

The data were retrieved from the online repository of the Romanian National Institute of Earth Physics (https://data.mendeley.com/datasets/28kv3gsgcz/2, accessed on 24 February 2024 [24]). For the most recent dates, the daily data updates can be found at http://geobs.infp.ro (last accessed on 24 Febreuary 2024). Once the data were collected, they were sorted using a Python script that arranges the input according to the station it belongs to and using subfolders for each month from January 2016 to December 2022. Since the signal’s seasonality must be removed, it was important to use as much data as possible. It should be highlighted that these indoor radon stations are not affected by precipitation, which was checked via the cross-correlation of humidity with precipitation, and we found no existing relationship between the signals.

The sampling period was 1 min for data earlier than 2022 and 3 h for the most recent entries (from 2022 onwards). Therefore, in order to eliminate the daily radon cycles, a 1-day period was imposed in the time series analysis so that the daily radon cycles [7,25] were eliminated. This means that for each day, the median radon concentration [Bq/m^3^] was calculated using 1440 daily samples.

In Figure 2 the four multi-parametric stations with continuous ^222^*Rn* signal records during the periods in which the relevant earthquakes occurred are shown. In the case of the MLR station, the signal record was not available from mid-2017 onward, and in the case of the PLRd2 station, no records were retrieved from 2016 to 2018. For this reason, these stations are not considered in this work. For the signal’s seasonality extraction process, only the records from 2016 to the end of 2020 were used as they did not contain major gaps that could affect the analysis.

## 3. Correlation Between Seismic Activity and Radon Emissions

### 3.1. Analysis of Radon Time Series

Once the data were selected, the next step was to deseason and detrend the signal to obtain the residual. The pre-processing of the signal was performed using a modified version of the software Environmental-WaveletTool from Galiana-Merino et al. [26].

The first step carried out in the pre-processing stage was gap filling. This was a crucial process because some further processing techniques could not be applied if the signal presented discontinuities or gaps.This is an ambiguous operation, as the original signal cannot be retrieved after this step. The objective is to complete the gaps following the low- and high-frequency behaviour of the signal. For this, the signals were modelled using cubic splines (Spline-based Segmentation Technique, [27]), obtaining the piece-wise polynomial that best fit the signal and the corresponding residuals. The cubic spline interpolation provided the low-frequency behaviour for the complete signal, including the gaps. Meanwhile, the standard deviation of the residuals in the periods without gaps provided an estimation of the high-frequency behaviour. The contribution of both parts allowed us to fill in the gaps following the tendency of the signal. If an anomaly had happened during a period with a gap, it would not be retrieved, as it would not be part of the normal behaviour of the signal. Figure 3 shows the signals with the gaps filled in. The parts of the signal that were repaired are shown in green. The colour code for the stars that mark the earthquakes has been maintained throughout the work to correlate the different figures.

In the following step, the linear and seasonal effects were also removed following the procedure explained in Galiana-Merino et al. [9]. That study shows that the radon series includes 1-year-long oscillations, which are due to other external factors, but cannot be attributed to the seismic activity of the area, so it is essential to first eliminate this seasonal behaviour. Thus, the 1-year oscillations, as well as any possible linear trend, were removed by estimating the best fit with the cubic spline.

Figure 4 shows the radon residual, from now on referred to as Δ^222^*Rn*, for the selected stations.

The signal is still noisy, but some features can be seen and pointed out. The median value is now close to zero for all the selected stations, unlike the non-deseasoned signals shown in Figure 3, as expected from a residual component. We refrain from applying a moving average filter, as this part of the smoothing will be discussed in the results when comparing the cross-correlation between the stations. Nonetheless, we refer the reader to the section Appendix A for the smoothed residuals using either a 15-day moving average filter or a 30-day moving average filter.

It can be seen that for the dates on which the major earthquakes occur, not all the stations show values much higher than the standard deviation σ, but, for instance, stations BISRd and NEHRd show a sharp increase almost a month before the 23 September 2016 earthquake, whereas the increase in the LOPRd barely surpasses the standard deviation of the median residual. Before the 8 February 2017, an increase in the radon residual above one σ can be seen for all the stations. In the case of the 2 August 2017 earthquake, only the LOPRd and NEHRd stations show a clear increase. In the October 2018 earthquake, three stations (BISRd, LOPRd and NEHRd) show an increase in the radon residual before it occurs, while VRID seems to increase after it has happened. For the most recent earthquake, the tendency is not so clear.

### 3.2. Seismic Activity and Radon Emissions

The observed behaviour of the radon signals described in Section 3.1 and the occurrence of major earthquakes in the area of study motivated us to examine the relationship between seismicity and radon emissions. There exists no general rule to select which earthquakes are responsible for the radon emissions registered at each station. Since all the earthquakes are related to the same seismic source, we followed the hypothesis from Hauksson and Goddard [28], which relates the magnitude of the earthquake to the maximum distance at which the radon anomaly caused by such an event can be detected (Equation 1): (1)$$M\geq 2.4\cdot \log_{10}D-0.43$$
where *M* is the magnitude (in Mw units), and *D* is the distance (in kilometres) from the earthquake location to the place where the anomaly is recorded.

In order to select the magnitude, which, in this case, was a lower bound (magnitude cut-off), the Depth–Magnitude distribution was examined for the seismic events in the Vrancea region. The formerly described catalogue was filtered so that the events with a recording quality of C or higher (there are four categories: D, C, B and A, from worst to best) were considered. Figure 5 shows the Depth–Magnitude distribution for the 10,942 events in the filtered catalogue, with magnitudes ranging from 0.0 to 7.7 Mw.

Two clearly defined clusters can be seen in Figure 5: one for shallow seismicity (from the surface down to 50 km) with magnitudes centred around 2.0 Mw, and another for intermediate-to-deep seismicity (50 km and below) with magnitudes ranging from 2.5 up to 7.7 Mw and centred around 3.0 Mw (with higher dispersion towards greater magnitudes). It can be noted that shallow seismicity (shallower than 50 km) accounts for approximately the 35.5% of the data, whereas intermediate-to-deep seismicity makes up the remaining 65.5%.

Given the seismotectonic context and the information from the Depth–Magnitude plot, the magnitude cut-off was set to 3.1 Mw. With this magnitude value, the maximum distance for the radon anomaly to be recorded can be calculated with (Equation 1) to be 30 km.

Regarding the parameters that represent the source-to-radon-station distance, it should be noted that different studies [29,30] conducted on the length diffusion of ^222^*Rn* have found ranges from 1.6 to 1.9 m in different samples of dry soils and from 0.7 to 1.5 cm in saturated clay samples. This means that the epicentre should be considered in distance computations rather than the hypocentre, as there is no physiochemical model that could explain the ^222^*Rn* anomalies if the noble gas was released at the depth at which the rupture happened. Herein, the aim of this work is not to explain the physiochemical phenomena behind the radon anomalies, a complex issue that has been extensively addressed [31,32,33]. However, for these anomalies to be able to be recorded, a process in which all of the Vrancea slab is involved should be considered. The hypothesis is that a common stress field is responsible for both the seismicity and the radon emission anomalies, hence the search for a relationship between both phenomena.

The subcatalogues for each of the stations were used to study whether there exists a spatio-temporal correlation between the $\Delta^{222}$*Rn* signal and the seismic activity rate. The multi-parametric stations near the area of study with available data and their influence areas are presented in Figure 6.

The seismic activity rate can be computed by first obtaining the seismic parameters following the Aki-Utsu [34,35]) Weighted Maximum Likelihood Estimation (MLE) (Equation 2): (2)$$\hat{b}=\frac{\log_{10}e}{\bar{M}+0.5\cdot \Delta M-M_{min}}$$
where $\hat{b}$ is the estimated *b*-value, $\bar{M}$ is the mean magnitude of the catalogue, ΔM is the binning of the catalogue (taken as 0.1) and $M_{min}$ is the completeness magnitude of the catalogue. Then, the Gutenberg–Richter law (G-R) [36] is used to obtain the *a* parameter and the daily seismic activity rate (Equation 3): (3)$$\lambda \left(m\geq M\right)=\frac{10^{a-\hat{b}\cdot M}}{\Delta t}$$
where λ(m≥M) is the daily seismic activity rate for magnitudes greater than or equal to *M*, *a* is the productivity parameter from the G-R law and Δt is the time span of the catalogue.

Using the catalogues for each of the stations, the daily seismic activity rate was computed for the location of the station. This was achieved by computing the *a*- and *b*-values for each day at each station by using the rolling window method with a minimum of 50 events per window. Figure 7 shows the seismic activity rate for a threshold magnitude of 3 Mw for the selected stations, as a result of applying (Equation 3) to the *a*- and *b*-values’ time series for each station.

### 3.3. Testing Correlation Between Seismic Activity Rate and Radon Emissions

The main hypothesis is that the stress field that is responsible for seismicity in the Vrancea region must also be responsible for the radon emission anomalies, i.e., the radon emission component that is solely dependent on the tectonic stress field. Given that this hypothesis holds true, a correlation should exist between these two phenomena.

If, as a result of this correlation experiment, a delay between both signals can be found, it could mean that one of the signals could be used to forecast the value of the other at a future time. For this to be achieved, a consistency in the delay of the signal must be found for different periods.

Before approaching the correlation between the radon residuals and the seismic activity rate, a cross-correlation study between the multi-parametric stations had to be conducted. For each of the earthquakes marked in Figure 4, which constitute the subset of the earthquakes in Table 1 that occurred during the period with continuous radon monitoring of the selected stations, a cross-correlation was conducted between the selected stations using the whole period and compared with that using only 4 months of radon data (2 months prior to the earthquake and 2 months after the earthquake). This duration for the analysis was selected based on the fact that radon anomalies can have durations in the range of months and even years, which ensured that the tendency of the signal could be captured at all of the stations. For all of the inter-station correlation experiments, the null hypothesis is that there exists no correlation between the radon measurements at each of the stations, i.e., the data reflect a local phenomenon that cannot be correlated with regional-scale seismicity.

In order to check whether a correlation exists between the seismic activity rate and the radon residual signal, Contracting Window Analysis was conducted with a fixed endpoint and variable size (minimum size: 30 days). This method is a modified version of the Rolling Window Analysis [37], which is more suited for time series with complex behaviour that can change over time, and it was used to optimise the correlation coefficient to obtain the lag between the signals. The procedure to obtain the lag between the signals can be seen in the following Algorithm 1.
**Algorithm 1** Obtaining the lag between the signals.
Input: ^222^*Rn* signal and λ(m≥3.0)Output: lag between signals and maximum correlation value 1: Correlation optimisation for lag search 2: **for** all the window sizes down to 30 days (advancing the start point) **do** 3:  compute lag (*l*) and maximum correlation between signals and store in lists 4: get maximum correlation value ($\rho_{max}$) and index ($arg(\rho_{max})$) from the correlation list 5: get lag using $arg(\rho_{max})$ as index of the lag list 6: **return**
$\rho_{max}$ and lag

## 4. Results

First, the cross-correlation between the radon signals recorded by the different analysed stations had to be computed in order to check whether the stations were registering regional phenomena to be studied, or whether they were capturing independent (non-correlated) signals. For this task, the Pearson correlation coefficient and the *p*-value were computed for the raw signals (without any pre-processing) along the common duration of the radon records (from 2016 to 2020). After that, a 15-day and a 30-day moving average filter was also applied to the signal, similarly to the work of Piersanti et al. [7], in order to check if it improved the correlation.

In Table 3, it can be seen that smoothing the signal greatly increases the correlation coefficient as well as the *p*-value, making it possible to reject the null hypothesis for the cross-correlation between stations. From this preliminary analysis, we can deduce the stations with maximum cross-correlation, i.e., the stations BISRd, LOPRd and NEHRd. Therefore, a correlation study was conducted between the radon residuals estimated at these stations and the seismic activity rate obtained at their corresponding locations.

The figures corresponding to the cross-correlation matrices for the raw signal, 15-day moving average smoothed signal and 30-day moving average smoothed signal can be found in the section Appendix B.

Although the Pearson correlation coefficients are not very high, it can be seen that considering shorter periods (instead of the complete records from 2016 to 2020) improves the correlation results, as shown in Figure 8, where a period of 4 months centred around 23 September 2016 (the date of the occurrence of the 5.5 Mw earthquake) was selected. This is mostly due to the nature of the changes in the radon residual, which may be due to several factors, making the changes over time non-predictable.

The first plot of the daily seismic activity rate equal to or greater than 3.0 Mw and the radon residual for the three correlated stations is shown in Figure 9. An increase in the radon residual signal before the increase in seismic activity can be seen for the three stations during the period in which the 23 September 2016 earthquake occurred.

Figure 10 shows the correlation between the annual seismic activity rate for magnitudes equal to or greater than 3 Mw and the radon residuals for the three cross-correlated multi-parametric stations. The radon residual is coloured according to the correlation coefficient value. A 15-day and 30-day moving average filter was applied to both signals (radon residual and seismic activity rate) in order to compare them with the non- smoothed signals.

It can be seen that applying the moving average filter increases the maximum correlation coefficient in most cases. BISRd and NEHRd show a high correlation for the smoothed signals. In the case of the LOPRd station, the signal should be considered with caution, as there exist several peaks that could lead to difficulty matching the signals. The BISRd station shows a peak in the radon residual days before the earthquake occurs and increased values for both the seismic activity rate and radon residual from almost a month before the earthquake until it takes place. Since the seismic activity rate shows an increasing tendency before the earthquake’s occurrence at all of the stations, it is important to be able to correlate both signals even if the lag is not negative (meaning the radon signal forecasts the seismic activity rate signal), as it could still be useful depending on the behaviour of the seismic activity rate in relation to the important earthquakes in the region.

For the rest of the earthquakes, the same procedure was applied. The figures related to each of the periods in which the relevant earthquakes happened for the area of study can be found in Appendix B. A summary of the results obtained for the BISRd, LOPRd and NEHRd stations is presented in Table 4, Table 5 and Table 6, respectively.

## 5. Discussion

A significant correlation (p<0.0001) is found between the smoothed radon residual signals for the duration of the records for the stations BISRd, LOPRd, and NEHRd, which improves greatly when shorter periods are considered (ρ≥0.57), as seen in Figure 7.

Table 4, Table 5 and Table 6 evidence that applying a smoothing filter to both signals (seismic activity rate and radon residual) considerably increases the correlation coefficient. Using a 15-day moving average seems to both increase the correlation coefficient (which remains higher than 0.6 with the exception of four instances, three of which are from the station LOPRd) and prevent the loss of information that can help with the correlation. For instance, in Figure 10, it can be seen for the BISRd station that using a 30-day moving average filter smooths a few peaks of the radon signal that could be correlated with the seismic activity rate. This can be also seen in Figure A4, for the period in which the 8 February 2017 earthquake occurred, at the station BISRd, where the shape of the peaks is smoothed by using the 30-day moving average filter in contrast to the 15-day moving average filter. In this sense, although the 30-day moving average increases the correlation coefficient in almost every case (when compared to the 15-day moving average filter), it can over-smooth the signal.

Even though the smoothing filters reduce the noise due to oscillations in both the seismic activity rate and radon residual signals, there exist several factors that can influence them and hinder the correlation. For this reason, although the correlation is performed through a Python script, it is important to check the results to see whether some peaks have been miscorrelated. This can be seen in Table 4, Table 5 and Table 6, where the lag computed for the signal with the 15-day moving average smoothing applied is very different from the one with the 30-day moving average. This is the case for the entries in a bold font in the tables. This happens for two periods, the one covering the earthquake from 2 August 2017 and the one from 31 January 2020. In the first period, Figure A8 reveals that for the BISRd station, two peaks that can be fitted to the signal are seen; the most probable value for lag in this instance would be 0 days instead of −30 or 29 days. This could be also the case for the period covering the earthquake in January 2020.

The most interesting results regarding the lag values’ spatial distribution and their relation to the seismicity in the region are presented in the following paragraphs.

Figure 11 shows that the stations southwest from the location of the epicentres exhibit higher lag values, i.e, the radon signal peaks before the seismic activity rate does for the LOPRd and NEHRd stations than it does for the BISRd station.

To see whether this spatial distribution is related to the evolution of seismicity with time, a cross-section spanning the study area in the SW-NE direction from the location of the NEHRd station towards the epicentre’s location is presented in Figure 12.

The time–space plot indicates migration of the epicentre’s location towards the SW of the BISRd location (in the projected A-A′ direction) in the first 50 days (since August 2016). Most of the earthquakes occurred at depths greater than 70 km. This is coherent with migration of the stress field towards the SW of the area of study, which could, in turn, explain radon residual peaks occurring before the increase in the seismic activity rate.

In the case of the period in which the 8 February 2017 earthquake occurred, it can be seen that the station which is closer to the epicentre has greater lag than the other stations, meaning the radon residual peaks before the seismic activity rate increases. This can be seen in Figure 13 and Figure 14.

Another example of this behaviour can be seen for the period in which the 28 October 2018 earthquake happened. Figure 15 shows the spatial distribution of the lag values.

Figure 16, on the other hand, shows the cross-section of the epicentre’s projected location from August 2018 until the occurrence of the earthquake. Again, although these are projected distances along the A-A′ trace, it can be seen that the epicentres are migrating towards the location of the main earthquake, which could explain the negative lag value for the LOPRd station.

For the rest of the earthquakes no clear conclusions can be drawn, as some of the lag values may be a result of a miscorrelation. The figures related to the corresponding cross-section plots can be seen in Appendix C.

## 6. Conclusions

In this work, the radon residual signal was computed for the multi-parametric stations in the Vrancea region by deseasoning and detrending the raw data. Using the seismic catalogue properties, a minimum magnitude of 3.0 Mw was selected with a corresponding rounded 30 km distance around the stations in order to create seismic catalogues for each one. With these catalogues, the seismic parameters were computed, and thus, we determined the seismic activity rate for each of the stations’ locations.

For three of the considered periods, a pattern of increasing lag is seen (the radon residual increasing/peaking before the increase in the seismic activity rate) towards the location of the nearest station to the epicentre (Figure 13 and Figure 15) or towards where the seismicity is increasing prior to the main earthquake in the period (Figure 11, as seen in the cross-section in Figure 13). For the rest of the periods (2 August 2017 and 31 January 2020), no clear trend is seen. This could be due to several factors: miscorrelation between the signals (which, as commented before, could be related to the presence of several peaks in a short time span), a low value for the radon residual (this could be important in the 31 January 2020 earthquake period—see Figure 4), and border effects on the signal detrending process (again, this could explain the results for the 31 January 2020 earthquake period), amongst other reasons.

These results could be explained if the regional tectonic stress field changes are identified as the common cause for both the seismic activity rate and the radon residual changes. These correlation results could enable the search for a regional function for seismic activity rate forecasting using the radon signal given that a complete analysis of a general correlation function is conducted. Regarding this correlation function, several parameters could be considered, for instance, the azimuth, the distance between stations and seismic activity rate grid points, $V_{S30}$, tectonic stress parameters, the focal mechanism of the nearest high-magnitude earthquakes, proximity to local faulting systems, etc.

## Figures and Tables

**Figure 1 sensors-25-00933-f001:**
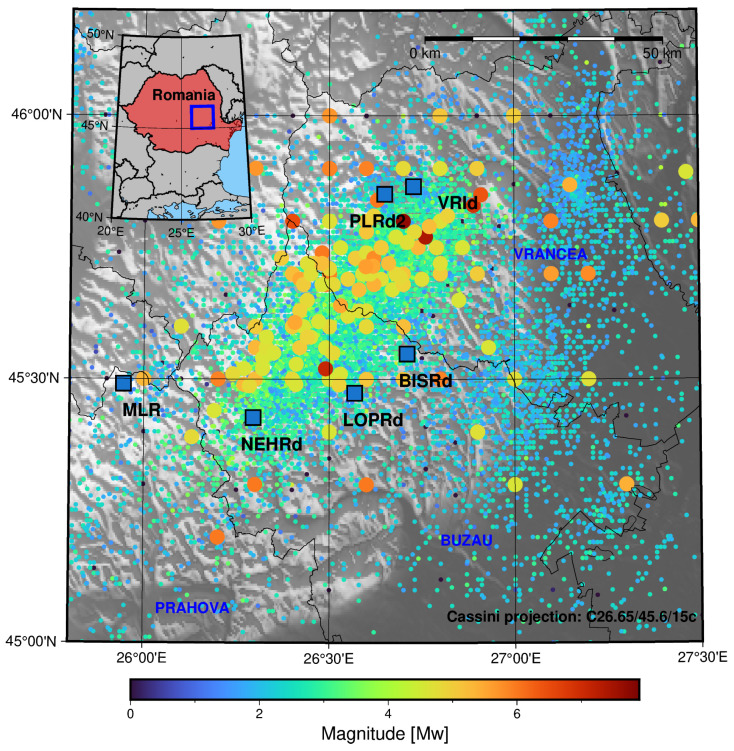
The ROMPLUScatalogue for the Vrancea region (Romania) from 1984 to 2023. The colour of the markers depends on the magnitude of the events. Regarding size, the events with magnitudes of less than 4.5 are represented by 0.1 cm markers, whereas a 0.2 cm diameter was chosen for events with greater magnitudes. The blue squares mark the location of the multi-parametric stations in the Vrancea region. The blue rectangle in the inset represents the area of study in the main plot, whereas the red polygon represents Romania.

**Figure 2 sensors-25-00933-f002:**
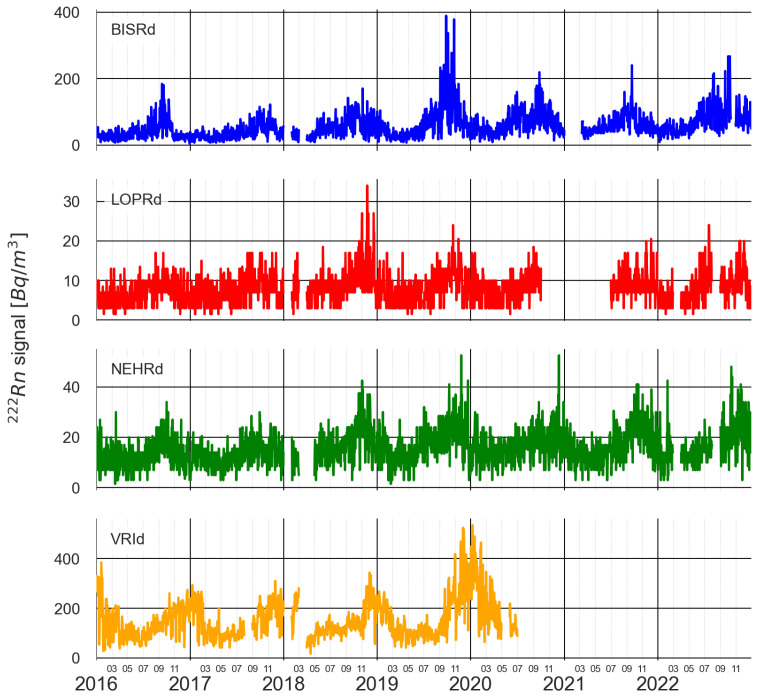
The ^222^*Rn* signal time series for the preselected multi-parametric stations located inside the area of study.

**Figure 3 sensors-25-00933-f003:**
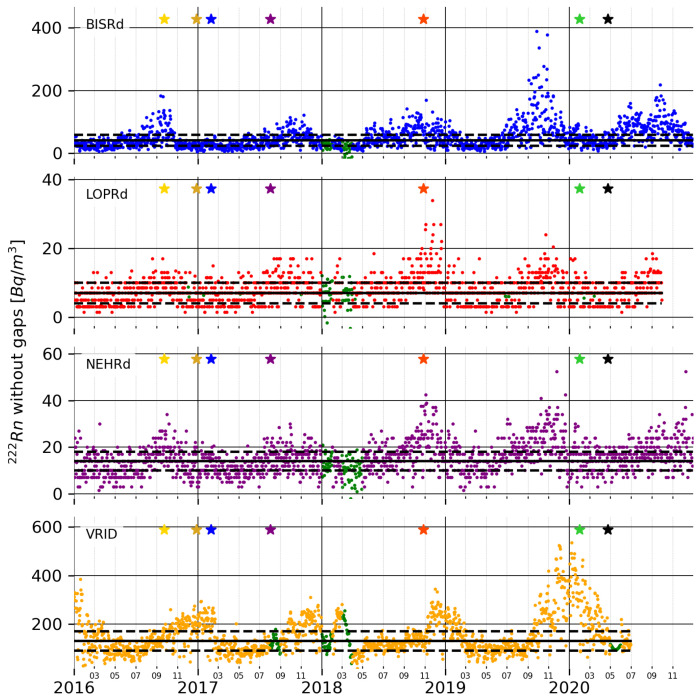
The ^222^*Rn* signal without gaps for the selected stations. The green sections of the data plot show the periods with gaps in the original signal. The dashed lines constrain the values within the plus/minus standard deviation. The stars mark the strong earthquakes listed in Table 1 that are within the period covered by the radon time series.

**Figure 4 sensors-25-00933-f004:**
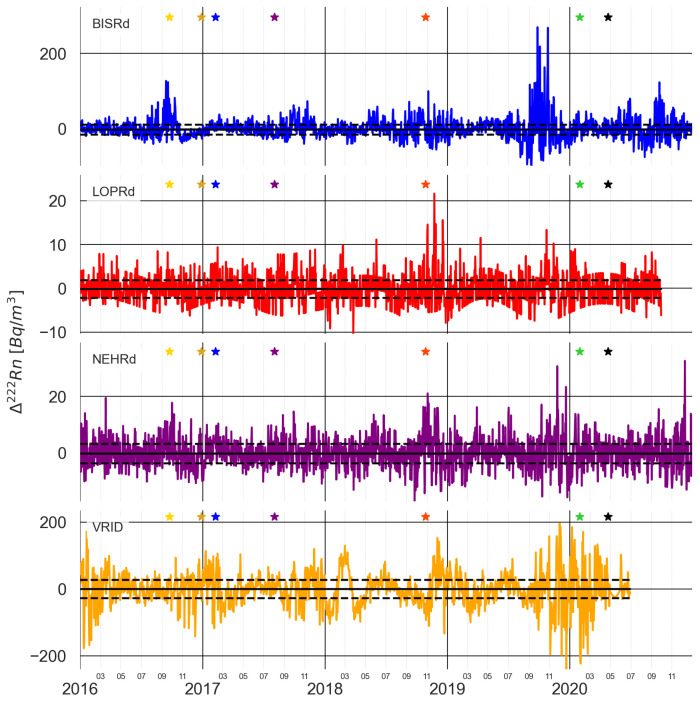
The $\Delta^{222}$*Rn* signal for the selected stations. The solid black line marks the median value and the dashed black lines constrain the values within the plus/minus standard deviation. The stars mark the strong earthquakes listed in Table 1 that are within the period covered by the radon time series.

**Figure 5 sensors-25-00933-f005:**
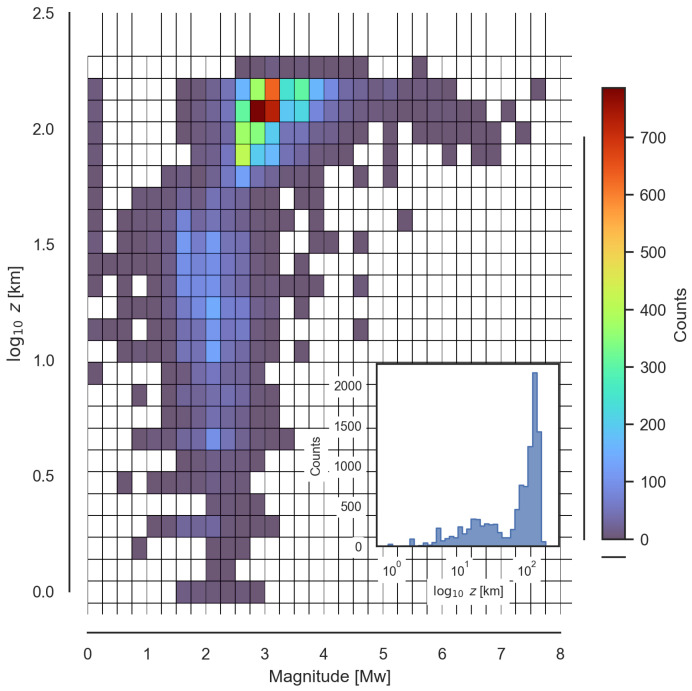
A 2D histogram plot for the Depth–Magnitude distribution of Vrancea’s seismic catalogue for events with quality better than or equal to C. The inset in the bottom right corner shows the number of earthquakes in each depth bin.

**Figure 6 sensors-25-00933-f006:**
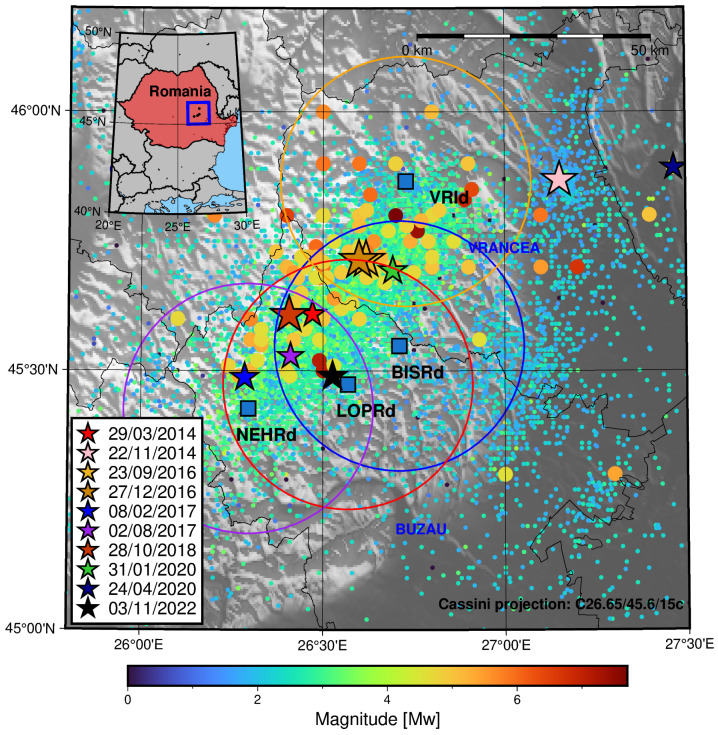
The circles around the blue squares (station locations) represent the 30 km influence area. The colour of the markers depends on the magnitude of the events. Regarding size, the events with magnitudes of less than 4.5 are represented by 0.1 cm markers, whereas a 0.2 cm diameter was chosen for events with greater magnitudes. The stars mark the earthquakes from Table 1.

**Figure 7 sensors-25-00933-f007:**
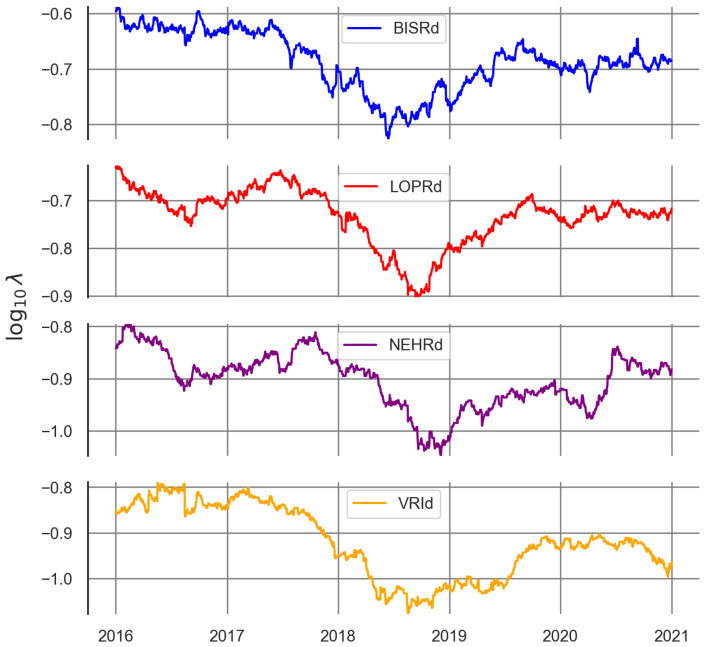
Seismic activity rate for magnitudes greater than or equal to the 3.0 Mw computed for the selected multi-parametric stations.

**Figure 8 sensors-25-00933-f008:**
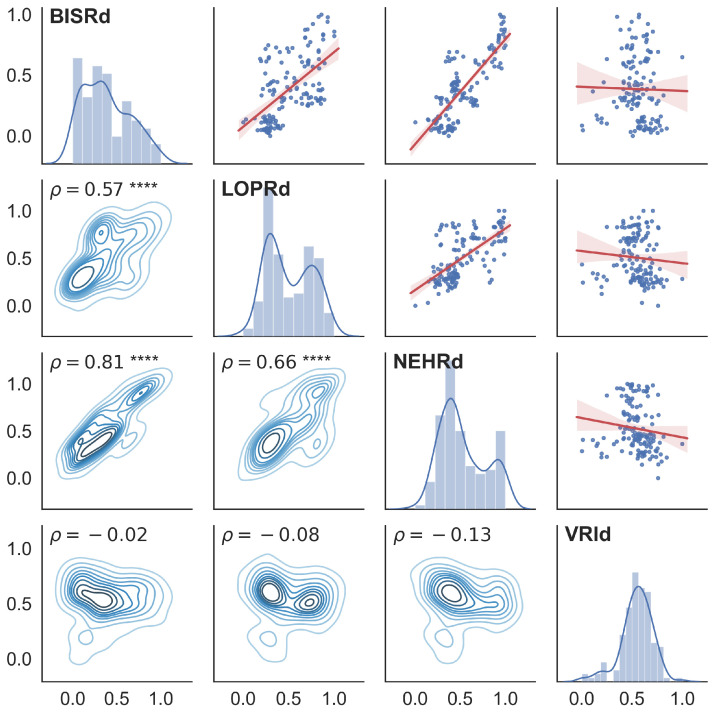
A cross-correlation matrix for the period from August 2016 to December 2016 with a 15-day moving average filter applied to the radon residuals. The upper triangular matrix represents the correlation plot for the stations where the solid red line is the fit for the correlation coefficient, the main diagonal represents the histogram and distribution for each of the stations, and the lower triangular matrix represents the Kernel Density Estimate (KDE) plot for the bivariate distributions. The asterisk notation in the correlation coefficient (ρ) indicates the *p*-value range, meaning that **** equals a *p*-value of less than 0.0001, and no asterisk means a *p*-value > 0.05.

**Figure 9 sensors-25-00933-f009:**
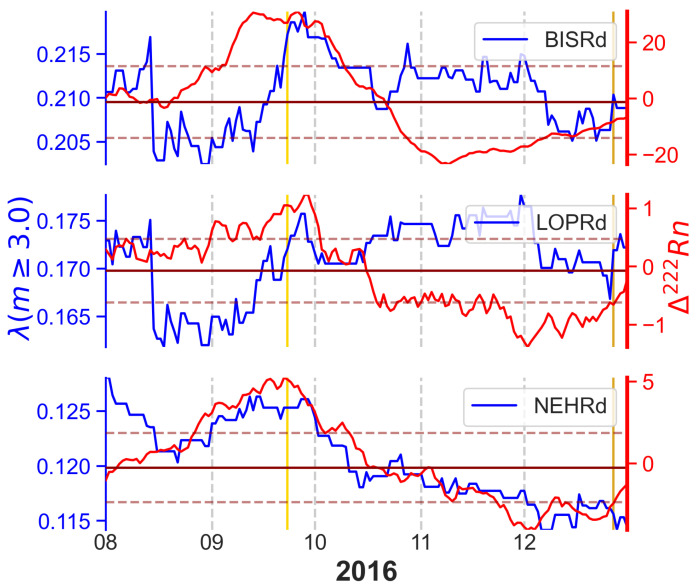
Radon signal compared with annual seismic activity. The solid red line marks the median value for the radon residual, and the dashed red lines constrain the standard deviation for the radon residual. The yellow line marks the date of the 23 September 2016 earthquake, and the golden line the 27 December 2016 earthquake.

**Figure 10 sensors-25-00933-f010:**
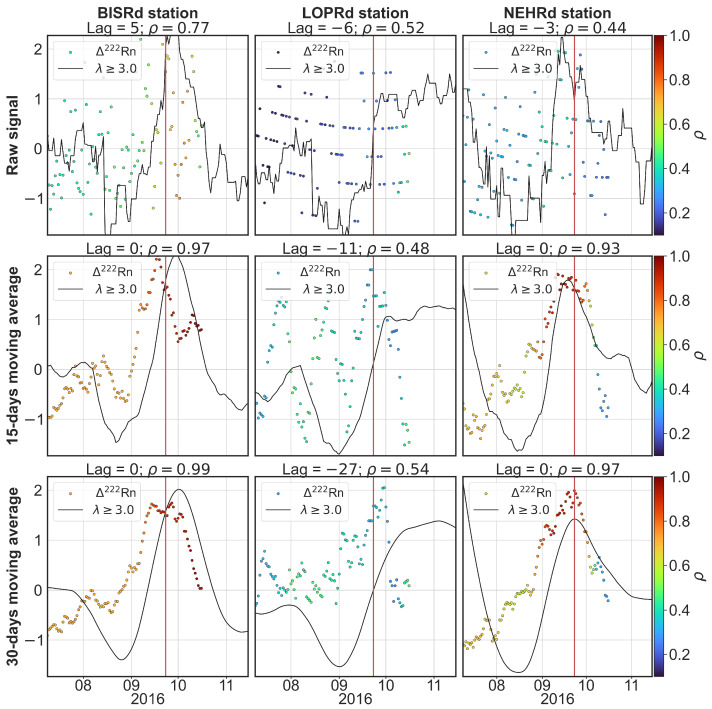
Foreach of the three stations, the radon residual signal is shown with coloured circles according to the maximum correlation value achieved during the correlation analysis that spans from that point to the end of the correlation window. The black lines represent the seismic activity rate for each of the station locations. The first row presents the raw (non-smoothed) signals, the second row shows both signals with a 15-day moving average filter applied, and the third row shows them with a 30-day moving average filter. The vertical red line marks the date of the earthquake, 23 September 2016.

**Figure 11 sensors-25-00933-f011:**
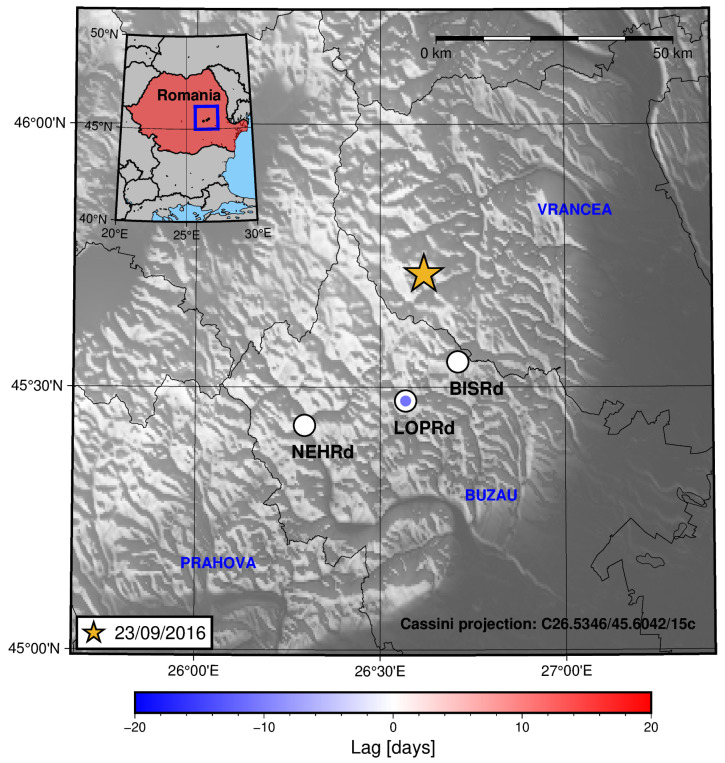
A map showing the lag between the radon residual and the seismic activity rate for each of the studied stations in the region for the period in which the 23 September 2016 earthquake occurred in Vrancea. The colour inside the circles is related to the lag value through the shown colour bar.

**Figure 12 sensors-25-00933-f012:**
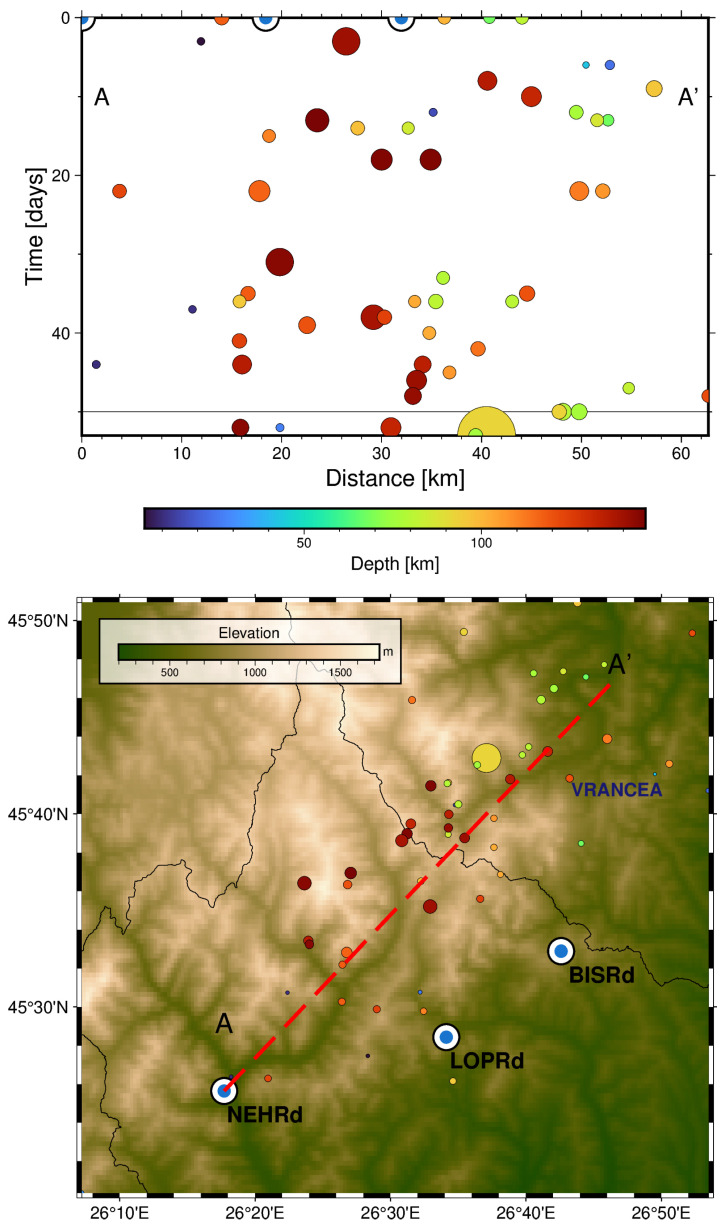
(**Top**): Time–Distance plot for the Mw ≥ 3 earthquakes from August 2016 until September 2016 projected along the A-A′ cross-section. The colour indicates the depth of the events and the size is proportional to the magnitude. (**Bottom**): Map with the epicentres, station location and cross-section trace.

**Figure 13 sensors-25-00933-f013:**
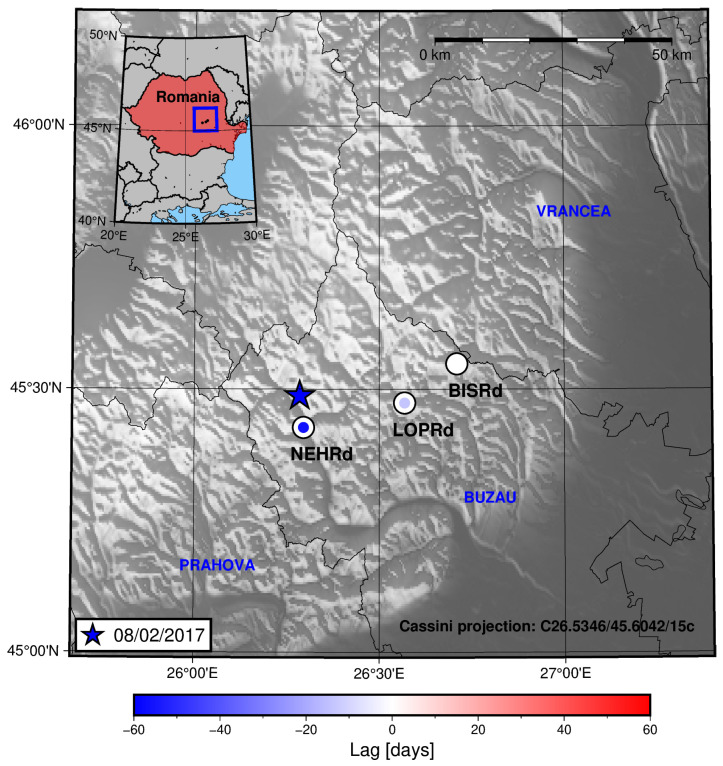
Map showing the lag between the radon residual and the seismic activity rate for each of the studied station in the region for the period in which the 8 February 2017 earthquake occurred in Vrancea. The colour inside the circles is related with the lag value through the shown colour bar. It can be seen that the closer to the epicentre location, the more negative the lag, meaning the radon peak is reached earlier than the seismic activity rate peak.

**Figure 14 sensors-25-00933-f014:**
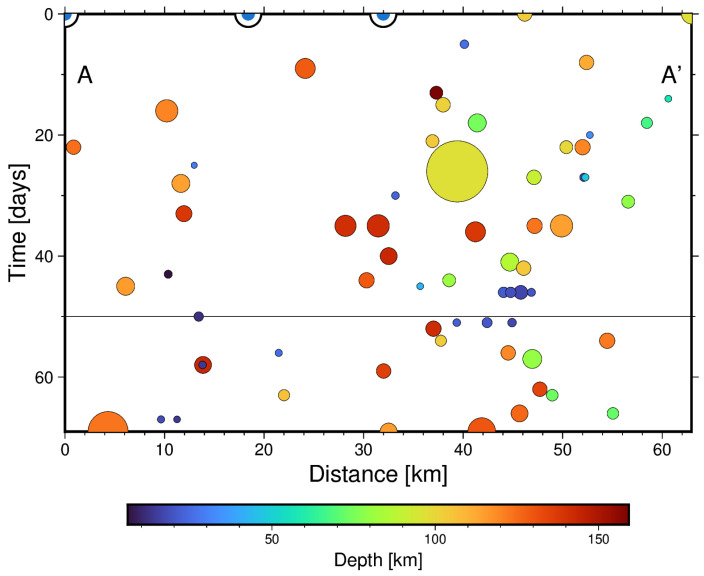
(**Top**): A Time–Distance plot for the Mw ≥ 3 earthquakes from December 2016 to February 2017 projected along the A-A′ cross-section. The colour indicates the depth of the events, and the size is proportional to the magnitude. (**Bottom**): A map with the epicentres, station location and cross-section trace.

**Figure 15 sensors-25-00933-f015:**
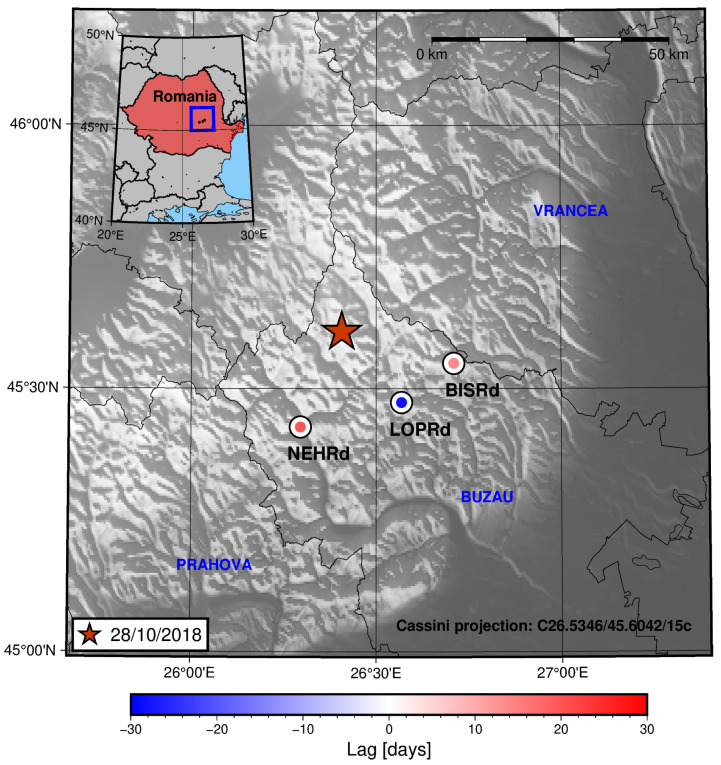
A map showing the lag between the radon residual and the seismic activity rate for each of the studied stations in the region for the period in which the 28 October 2018 earthquake occurred in Vrancea. The colour inside the circles is related to the lag value through the shown colour bar.

**Figure 16 sensors-25-00933-f016:**
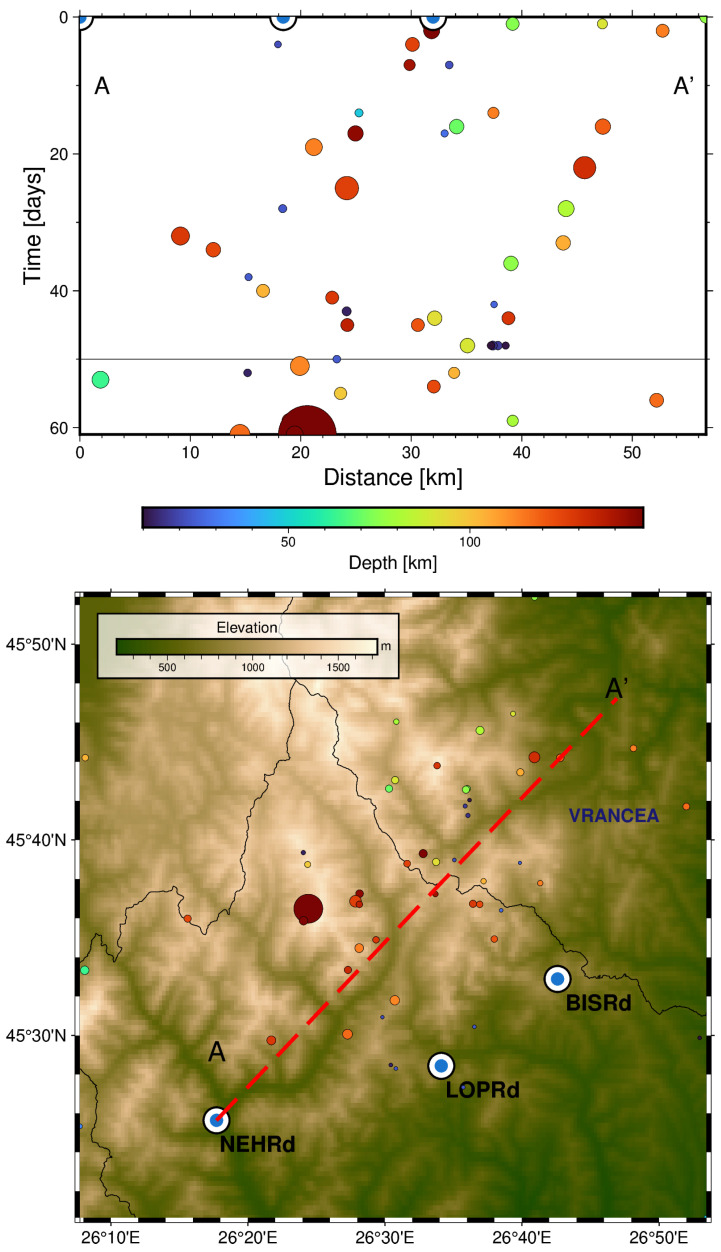
(**Top**): A Time–Distance plot for the Mw ≥ 3 earthquakes from August 2018 to October 2018 projected along the A-A′ cross-section. The colour indicates the depth of the events, and the size is proportional to the magnitude. (**Bottom**): A map with the epicentres, station location and cross-section trace.

**Table 1 sensors-25-00933-t001:** The most intense earthquakes in the area of study during the last 10 years.

Date [dd/mm/yyyy]	Time UTC [hh:mm:ss]	Magnitude [Mw]	Longitude [°E]	Latitude [°N]	Depth km
29/03/2014	19:18:05	4.6	26.4709	45.6094	134.4
22/11/2014	19:14:17	5.4	27.1517	45.8683	40.9
23/09/2016	23:23:11	5.5	26.6181	45.7148	92.0
27/12/2016	23:20:56	5.6	26.5987	45.7139	96.9
08/02/2017	15:08:21	4.8	26.2849	45.4874	123.2
02/08/2017	15:08:21	4.6	26.4106	45.5286	131.0
28/10/2018	00:38:11	5.5	26.4068	45.6079	147.8
31/01/2020	01:26:46	4.8	26.6918	45.6937	118.2
24/04/2020	22:04:19	4.6	27.4674	45.8918	22.6
03/11/2022	04:50:26	4.9	26.5262	45.4895	149.0

**Table 2 sensors-25-00933-t002:** Names and locations of the multi-parametric stations considered in this work.

Station Code	Location	Longitude [°E]	Latitude [°N]	Elevation [m]
BISRd	Bisoca	26.7099	45.5481	823
LOPRd	Lopatari	26.5686	45.4738	645
MLR	MunteleRosu	25.9450	45.4909	1370
NEHRd	Nehoiu	26.2952	45.4272	513
PLRd2	Plostina	26.6497	45.8513	617
VRId	Vrancioaia	26.7277	45.8657	424

**Table 3 sensors-25-00933-t003:** Cross-correlation between radon stations using the raw residuals, and the residuals after applying 15-day and 30-day moving average filters.

Stations	ρ Raw Signals	ρ 15-Day Moving Average	ρ 30-Day Moving Average
BISRd-LOPRd	0.06 *^,1^	0.38 ****	0.51 ****
BISRd-NEHRd	0.08 **	0.38 ****	0.44 ****
BISRd-VRID	0.09 ***	0.03	−0.00
LOPRd-NEHRd	0.21 ****	0.54 ****	0.64 ****
LOPRd-VRID	0.08 ***	0.08 **	0.04
NEHRd-VRID	0.14 ****	0.06 *	0.03

^1^ * *p*-value < 0.05; ** *p*-value < 0.01; *** *p*-value < 0.001, **** *p*-value < 0.0001, etc. No asterisk means *p*-value > 0.05.

**Table 4 sensors-25-00933-t004:** The correlation results for the periods in which the earthquakes occurred for the station BISRd. The entries in a bold font represent inconsistencies to be analysed in the discussion.

Date	ρ	Lag (Days)	ρ	Lag (Days)	ρ	Lag (Days)
	Raw Signals		15-Day Average		30-Day Average	
23 September 2016	0.77	5	0.97	0	0.99	0
27 December 2016	0.70	5	0.87	3	0.95	3
8 February 2017	0.44	7	0.85	0	0.93	0
2 August 2017	0.52	−28	0.65	**−30**	0.68	**29**
28 October 2018	0.36	2	0.70	14	0.66	12
31 January 2020	0.50	−1	0.87	0	0.95	0

**Table 5 sensors-25-00933-t005:** The correlation results for the periods in which the earthquakes occurred for the station LOPRd. The entries in a bold font represent inconsistencies to be analysed in the discussion.

Date	ρ	Lag (Days)	ρ	Lag (Days)	ρ	Lag (Days)
	Raw Signals		15-Day Average		30-Day Average	
23 September 2016	0.52	−6	0.48	−11	0.54	−27
27 December 2016	0.44	−16	0.74	−14	0.82	−10
8 February 2017	0.35	−22	0.58	−45	0.62	−40
2 August 2017	0.45	8	0.71	**10**	0.88	**0**
28 October 2018	0.36	−8	0.56	−28	0.77	−19
31 January /2020	0.38	5	0.62	**6**	0.68	**−51**

**Table 6 sensors-25-00933-t006:** The correlation results for the periods in which the earthquakes occurred for the station NEHRd. The entries in a bold font represent inconsistencies to be analysed in the discussion.

Date	ρ	Lag (Days)	ρ	Lag (Days)	ρ	Lag (Days)
	Raw Signals		15-Day Average		30-Day Average	
23 September 2016	0.44	−3	0.93	0	0.97	0
27 December 2016	0.46	24	0.79	25	0.87	13
8 February 2017	0.37	−55	0.76	−56	0.79	−53
2 August 2017	0.48	1	0.80	**−1**	0.72	**−14**
28 October 2018	0.42	23	0.70	20	0.56	17
31 January 2020	0.41	5	0.58	−49	0.61	−41

## Data Availability

All data included in this manuscript are available upon request by contacting the corresponding author.

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
