# Peer review of "Spatio-Temporal Correlation Between Radon Emissions and Seismic Activity: An Example Based on the Vrancea Region (Romania)"

_sensors, 2025, doi:10.3390/s25030933_

Round 1

Reviewer 1 Report

Comments and Suggestions for Authors

This research investigates the use of 222Rn as a tool for monitoring seismic regions, addressing an important gap in our understanding of soil gas emissions as potential earthquake precursors. The study’s originality and relevance are underscored by its contribution to earthquake surveillance methods. By correlating soil gas emissions with seismicity, this work offers valuable insights into seismic precursors, thereby enriching the existing literature.

The following points should be addressed:

Lines 82-89: this information does not add substance to the manuscript.

Lines 97-98: the earthquake concentration in this area is probably due to structural active lineaments (faults). It would be interesting to add some geology to explain seismicity of the area.

Caption of Figure 12: Ir should be It.

Some details on the precision and accuracy of the 222Rn instrument used for measurements, including any potential sources of error, would enhance the radon monitoring section.

Conclusions are generally consistent with the data but the summary in lines 344-355 should be avoided.

The references are well-chosen, though including more recent studies could better reflect the current research landscape. Piersanti et al is repeated (ref 7 and 26)

Author Response

Comments 1: This research investigates the use of 222Rn as a tool for monitoring seismic regions, addressing an important gap in our understanding of soil gas emissions as potential earthquake precursors. The study’s originality and relevance are underscored by its contribution to earthquake surveillance methods. By correlating soil gas emissions with seismicity, this work offers valuable insights into seismic precursors, thereby enriching the existing literature.

The following points should be addressed:

Lines 82-89: this information does not add substance to the manuscript.

Lines 97-98: the earthquake concentration in this area is probably due to structural active lineaments (faults). It would be interesting to add some geology to explain seismicity of the area.

Response 1: Thank you for pointing this out. We have revised this part of the manuscript and since both questions are related, we have decided to update the manuscript accordingly.

The blue font represents the modified version of the manuscript. Regarding the lines 82-89 we have changed their location and adapted the content of the paragraph to add to the explanation of the seismicity in Vrancea. This region has two ranges of seismicity in terms of depth, as discussed from Figure 4 in the depth-magnitude plot. The most important when it comes to magnitude is associated to the intermediate-deep part of catalogue (from 60 km to 200 km) that is why we think keeping this above-mentioned lines but changing its location can give the reader an idea of the complexity of the Vrancea seismic nest.

The new version of the manuscript has now a paragraph under the Figure 1 with part of the information that was in the lines 82-89:

It can be seen that the most intense earthquakes are concentrated around the eastern-most part of the Vrancea region towards the north-eastern limit of Buzau region. Regarding the origin of seismicity, one of the hypotheses is based on the presence of a subducted oceanic slab. Nevertheless, Knapp et al. 2005 [20] argued that a delamination process could explain the seismicity in the foreland of the Carpatian system not accounted for in earlier models for the subducted oceanic slab scenario. More recently, Müller et al. 2010 [21] pointed towards a non-coupled (or detached) slab scenario (with respect to the upper crust). In contrast to this model, Petrescu et al. 2021 [22] proposed a weakly-coupled slab scenario for Vrancea compatible with the observed foreland deformation. In terms of mechanisms that explain the physiochemical processes involved in the seismicity, Ferrand and Manea (2021) [23] found correlation between the intermediate-deep seismicity of the Vrancea region and the dehydration of the minerals in the subducted oceanic slab in this region.

New reference:

[20] Knapp. J.H.; Knapp, C.C.; Raileanu, V.; Matenco, L.; Mocanu, V.; Dinu, C. Crustal constraints on the origin of mantle seismicity in the Vrancea Zone, Romania: The case for active continental lithospheric delamination, Tectonophysics 2005, 410, 311-323. DOI: 10.1016/j.tecto.2005.02.02

Comments 2: Caption of Figure 12: Ir should be It.

Some details on the precision and accuracy of the 222Rn instrument used for measurements, including any potential sources of error, would enhance the radon monitoring section.

Response 2: We agree this information should be added to complement the radon monitoring section. Now in the section 2.3 this paragraph has been added:

The Radon Monitoring Network is composed of Radon Scout Plus stations (https://sarad.de/product-detail.php?lang=en_US&cat_ID=2&p_ID=37), which  provide relatively accurate and precise radon measurements, with typical instrumental error lower than 6%, depending on environmental factors and proper use. Each data point (for each minute) has been measured over integration periods of 3 hours for which the statistical error is around ±20%. Regular calibration and appropriate measurement conditions are essential to minimize sources of error, including environmental interference, sensor drift, and the influence of radon decay products.

More details about the radon stations and the monitoring network can be found in the references [8,24,25].”

Comments 3: Conclusions are generally consistent with the data but the summary in lines 344-355 should be avoided.

The references are well-chosen, though including more recent studies could better reflect the current research landscape. Piersanti et al is repeated (ref 7 and 26).

Response 3: We appreciate the comments and have corrected the references. We agree with the reviewer and lines 344-355 have been removed in order to be clearer in the conclusions.

Reviewer 2 Report

Comments and Suggestions for Authors

The article presents the results of the analysis of data from 6 radon measurement stations in Romania with a time step of 1 minute for the time interval 2016-2022 and 3 hours after 2022. For joint processing, the time series were reduced to the same time step of 1 day. Comments on the article:

1. As can be seen from the graphs in Figure 2, the time series contain large gaps in registration. In particular, 2 stations named MLR and PLRd2 contain data of such quality that it is likely that they cannot be included in the joint processing. For this reason, data processing was actually continued only for the remaining 4 stations (see Figure 3). For this reason, stations MLR and PLRd2 can be excluded from consideration and from the text of the article as substandard.

2. The data were pre-processed to eliminate seasonal variations. In this case, the data gaps in the remaining 4 time series were pre-filled (Figure 3). It is known that filling in missing data, especially when the gap duration is significant (i.e. not 1-3 readings), is an ambiguous operation, the application of which requires explanations, which are not provided in the article. Apparently, when filling in the data gaps, some considerations were used based on the use of the periodically correlated structure of these time series. These considerations should be provided in the article.

3. The comments to the graphs in Figure 3 present considerations regarding the type of precursors in radon behavior in the simplest form as amplitude anomalies exceeding a certain level of variations in the form of a value multiple of the standard deviation. However, the behavior of the time series in Figure 3 clearly demonstrates seasonality. In this regard, the question arises: why was there talk of eliminating seasonal effects before? If seasonal effects were removed (by what method, by the way?), then it was necessary to provide graphs of the behavior of the time series after removing seasonal trends. But these graphs are not in the article.

4. The abstract mentions the operation of "noise suppression". What did it consist of? There is no description of this operation in the text of the article. It is necessary to outline it and provide the appearance of the data both after "removal of seasonal trends" and "noise suppression".

5. Section 3.3 of the article presents the results of calculating the correlations between the radon time series and the intensity of the seismic process. Apparently, the correlations are calculated after the operations of "removing seasonal trends" and "noise suppression". That is, I return to the previous remark that these operations are simply mentioned in the article as if they are "well-known" and do not need to be described. But this is not so. These operations must be presented and substantiated. In addition, it is necessary to provide graphs of the change in the intensity of the seismic process with which the radon time series are correlated. Otherwise, the legitimacy of using the Pearson correlation coefficient is unclear. If the correlated time series differ greatly in their behavior from Gaussian processes, then using Spearman's rank correlations will be more adequate.

Author Response

Comments 1: The article presents the results of the analysis of data from 6 radon measurement stations in Romania with a time step of 1 minute for the time interval 2016-2022 and 3 hours after 2022. For joint processing, the time series were reduced to the same time step of 1 day. Comments on the article:

1. As can be seen from the graphs in Figure 2, the time series contain large gaps in registration. In particular, 2 stations named MLR and PLRd2 contain data of such quality that it is likely that they cannot be included in the joint processing. For this reason, data processing was actually continued only for the remaining 4 stations (see Figure 3). For this reason, stations MLR and PLRd2 can be excluded from consideration and from the text of the article as substandard.

Response 1: Thank you for your comment. We agree with that keeping the stations in Figure 2 while having such gaps in the time series is not beneficial towards the data analysis as it makes the plot less readable in terms of subplot size. We have changed the Figure 2 so it will only show the 4 stations with more available data, and we have modified the paragraph including lines 135-140 accordingly to reflect this change.

Previously:

In Figure 2 it can be seen that only 4 out of the 6 multi-parametric stations have continuous 222Rn signal records during the periods in which the relevant earthquakes occurred. This fact constraints the data to the following multi-parametric stations: BISRd, LOPRd, NEHRd and VRId. For the signal’s seasonality extraction process only the records from 2016 until the end of 2020 will be used as they do not contain major gaps that could affect the analysis.”

Now:

In Figure 2 the 4 multi-parametric stations with continuous 222Rn signal records during the periods in which the relevant earthquakes occurred are shown. In the case of the MLR station the signal record was not available from mid 2017 on, and in the case of the PLRd2 station no record was retrieved from 2016 to 2018. For this reason, these stations are not considered in this work. For the signal’s seasonality extraction process only the records from 2016 until the end of 2020 will be used as they do not contain major gaps that could affect the analysis.”

Figure 2 has been updated:

Comments 2: The data were pre-processed to eliminate seasonal variations. In this case, the data gaps in the remaining 4 time series were pre-filled (Figure 3). It is known that filling in missing data, especially when the gap duration is significant (i.e. not 1-3 readings), is an ambiguous operation, the application of which requires explanations, which are not provided in the article. Apparently, when filling in the data gaps, some considerations were used based on the use of the periodically correlated structure of these time series. These considerations should be provided in the article.

Response 2: Agree. We have modified this part of the manuscript to add information to the gap-filling process. It should be noted as well that no period with missing data has been considered in the seismic activity rate – radon residual correlation. We added a sentence making this clear, although none of the considered earthquakes happened in a period where data were missing from the radon signal record. We also added an additional figure in between the former Figure 2 and Figure 3 (now Figure 4) to perform a step-by-step signal transformation walkthrough.

New lines 149-165:

Once the data has been selected, the next step is to deseason and detrend the signal to obtain the residual. The pre-processing of signal is done using a modified version of the software Environmental-WaveletTool from Galiana-Merino et al. [28].

The first step carried out in the pre-processing stage is the gap filling. This is a crucial process because some further processing techniques could not be applied in case the signal presents some discontinuities or gaps. This is an ambiguous operation as the original signal cannot be retrieved after this step. The objective is to complete the gaps following the low and high frequency behaviour of the signal. For that, the signals have been modelled using cubic splines (Spline-based Segmentation Technique, [29]), obtaining the piece-wise polynomial that best fit to the signal and the corresponding residuals. The cubic splines interpolation provides the low frequency behaviour for the complete signal, including the gaps. Meanwhile, the standard deviation of the residuals at the periods without gaps provides an estimation of the high frequency behaviour. The contribution of both parts allows to complete the gaps following the tendency of the signal. If some anomaly had happened during a period with a gap, it is not retrieved as it would not be part of the normal behaviour of the signal. Figure 3 shows the signals with the gaps filled in. The parts of the signal that have been repaired are shown in green colour.”

Comments 3: The comments to the graphs in Figure 3 present considerations regarding the type of precursors in radon behavior in the simplest form as amplitude anomalies exceeding a certain level of variations in the form of a value multiple of the standard deviation. However, the behavior of the time series in Figure 3 clearly demonstrates seasonality. In this regard, the question arises: why was there talk of eliminating seasonal effects before? If seasonal effects were removed (by what method, by the way?), then it was necessary to provide graphs of the behavior of the time series after removing seasonal trends. But these graphs are not in the article.

Response 3: We thank the reviewer for pointing this out. The detrending of the signal has been done from the former Figure 2 to the Figure 3 (when we plot the residual of the radon signal, noted as D222Rn). In order to be clearer the caption of the Figure 4 (3 in the first version of the manuscript) has been changed and also the paragraph preceding such has been modified.

In the following step, the linear and seasonal effects are also removed following the procedure explained in Galiana-Merino et al. [9]. That study shows that the radon series includes 1-year-long oscillations, which are due to other external factors, but cannot be attributed to the seismic activity of the area, so it is essential to first eliminate this seasonal behaviour. Thus, the 1-year oscillations, as well as any possible linear trend, have been removed by estimating the best fitting with cubic-spline.”

Although it is possible to use a function f(x) = ax+ bsin(x/T + c) to fit this signal, we found that the cubic spline worked better (it works like a second order approach to the sine function with more freedom in the parameter optimisation). We present the Figure 4, without a smoothing filter applied, as the analysis on which filter we select (between a 15-days moving average and 30-days moving average) is discussed in the “Results” section when we approach the cross-correlation between the multiparametric stations.

To complete the analysis, we add the following paragraph to emphasize the fact that the smoothing of the signal will be deal with later. Nonetheless, we added a section in the Appendix (Appendix A, so the rest will be B, C and D) to show how the smoothing signal would look like when considering the full time series (as in the results and discussion only the periods of interest are shown).

The signal is still noisy, but some features can be seen and pointed out. It can be seen that the median value is now close to zero for all the selected stations, as opposed to the non-deseasoned signals shown in Figure 3. We refrain from applying a moving average filter as this part of the smoothing will be discussed in the results when comparing the cross-correlation between the stations. Nonetheless, we refer the reader to the section Appendix A for the smoothed residuals using either a 15-days moving average filter or a 30-days moving average filter.

Comments 4: The abstract mentions the operation of "noise suppression". What did it consist of? There is no description of this operation in the text of the article. It is necessary to outline it and provide the appearance of the data both after "removal of seasonal trends" and "noise suppression".

Response 4: Agreed, the abstract refers to a denoise process which is not accurate for what is done in this work. It is a mistake (most probably in translation) and it has now this sentence in the Abstract has been changed:

First, the recorded radon signals are preprocessed (detrended, deseasoned and smoothed)

We hope that this changes along with the ones made in the previous questions help to clarify the preprocessing section.

Comments 5: Section 3.3 of the article presents the results of calculating the correlations between the radon time series and the intensity of the seismic process. Apparently, the correlations are calculated after the operations of "removing seasonal trends" and "noise suppression". That is, I return to the previous remark that these operations are simply mentioned in the article as if they are "well-known" and do not need to be described. But this is not so. These operations must be presented and substantiated. In addition, it is necessary to provide graphs of the change in the intensity of the seismic process with which the radon time series are correlated. Otherwise, the legitimacy of using the Pearson correlation coefficient is unclear. If the correlated time series differ greatly in their behavior from Gaussian processes, then using Spearman's rank correlations will be more adequate.

Response 5: Following the previous questions, we have an additional figure to showcase the effect of both the deseason of the signal and the gap-filling process. We added a more thorough explanation in the deseason process and cited the corresponding work upon this methodology is based (Galiana-Merino et al. (2022) and Galiana-Merino et al. 2014). Regarding the changes in the seismic intensity for each of the periods in which the correlation study between the radon residual and the seismic activity rate is to be conducted the corresponding figures show the changes in both variables. In this regard, we have modified the figures in order for them to be clear (e.g. Figure 10 in the new version of the manuscript).

The black lines represent the daily seismic activity rate for each of the stations using different smoothing values (i.e. 15-days moving average or 30-days moving average).

Round 2

Reviewer 2 Report

Comments and Suggestions for Authors

I suppose that the authors made a big work to improve the text and now it can be published in a current form.